# SKETCHING FASTER THAN DIMENSION TIMES UPDATE TIME

## ABSTRACT

In situations such as distributed computation, where one is interested in applying a sketch to a fixed vector, it is often possible to apply the sketch with a runtime that is faster than simulating the corresponding streaming algorithm. For the settings we consider, this avoids an $\omega(1)$ update time lower bound by [Larsen, Nelson, Nguyen '15] when the sketching algorithm has access to the entire vector. We consider a variety of problems.

- For the $\ell_2$ heavy-hitters problem, we give a space-optimal sketch which can be applied in linear time (with no logarithmic overhead). We also combine with the ExpanderSketch of [Larsen, Nelson, Nguyen, Thorup'16] to achieve fast decoding time, as well as with a tensor sketch.
- For $\ell_p$ estimation with $p \geq 2$ we apply our heavy-hitters scheme to give a linear-time sketch with dimension $\tilde{O}(d^{1-2/p})$, which is nearly optimal.
- Using ideas similar to our $\ell_2$-heavy-hitters sketch, we also address linear regression and low-rank approximation, and give sketches that are linear-time in natural regimes.
- Finally we introduce a reshaping trick and apply fast matrix multiplication algorithms to speed up $\ell_p$ approxiamtion for $1 \leq p \leq 2$.

We discuss applications of these techniques to distributed algorithms.

## 1  INTRODUCTION

As data becomes larger, it becomes increasingly important to develop algorithms that scale to massive datasets. One approach for dealing with such big data is to compress it into some smaller data structure from which useful statistics can still be calculated. This is the central idea of sketching. Formally, a (linear) sketch takes as input a data vector $v \in \mathbb{R}^d$, and compresses it via a linear map $S \in \mathbb{R}^{m \times d}$, to produce a vector $Sv \in \mathbb{R}^m$. Typically the goal is for $m$ to be dramatically smaller than $d$, so that storing $Sv$ is much more efficient than storing $v$ itself.

Historically, sketching was developed as a technique in the *streaming* setting where the goal is to process a huge number of updates. Here the model is that there is an underlying vector $v \in \mathbb{R}^d$ that is updated coordinate-wise. That is, one receives updates $u_i$ each with a single non-zero entry which modifies $v$ by $v \leftarrow v + u_i$. When the updates $u_i$ are unrestricted, this is typically referred to as turnstile streaming, in contrast with streams that only allow for non-negative updates.

In the streaming setting one is usually most interested in the space requirements, but other features such as the time to process updates are also interesting. To maintain a linear sketch $Sv$ of $v$, one needs to compute $Su_i$ for each update. Since the updates may be numerous, it is often important to design $S$ so that $Su_i$ can be efficiently calculated, while maintaining low storage requirements Alman & Yu (2020); Kane et al. (2011); Chou et al. (2019).

Sketches have also found important applications outside of streaming algorithms. For example a fast sketch can be applied as a preprocessing step to the reduce the size of a large scale optimization or machine learning problem so that the resulting problem can be solved more quickly Woodruff et al. (2014). More relevant to our setting, sketches have also found applications in improving the *communication complexity* of various problems Kannan et al. (2014); Woodruff & Zhang (2013); Huang et al. (2021). The typical setup is when one's data is partitioned among two or more servers

and the goal is to compute some quantity related to the underlying data, while minimizing the total communication among the servers Roughgarden et al. (2016). Sketches are useful here as the servers may first compress their data before sending it over the network, thus reducing the number of bits that need to be sent. As long as the sketches are linear, the sketches can then be aggregated and decoded by a "coordinator" who receives all the sketches.

While the sketches used for streaming algorithms and communication are often similar, the specific features that we want are different in these settings. For example, suppose one wants to apply a sketch $S$, which has been optimized for a streaming algorithm, now in a communication setting. If $v \in \mathbb{R}^d$ is the vector to be sketched by one of the servers, then the server could just feed the streaming updates one coordinate of $v$ at a time into the sketch, resulting in a runtime of $d \cdot$ (update time). Sometimes this is optimal. For instance if $S$ is a CountSketch Charikar et al. (2002), then each update requires $O(1)$ time, resulting in $O(d)$ time to form the sketch. This is not always the case though. For example, if one is interested in an $\ell_2$ heavy-hitters sketch with the optimal $O(\log(d)/\epsilon^2)$ dimension and high $O(1/\text{poly}d)$ failure probability, then all known sketches require $\omega(1)$ update time. The same is true for related problems, such as $\ell_p$ estimation. In fact, Larsen et al. (2015) even shows that such problems require $\omega(1)$ worst-case update time for turnstile streaming algorithms[1]!

In our setting, we have access to $v$ directly, and so there is no reason that we need to use the streaming updates to compute the sketch of $v$. In this situation, we ask whether it possible to sketch $v$ in truly linear time. A natural question is, when given full access to a vector, is it possible to apply a sketch in truly linear time without log factors, while simultaneously achieving an optimal sketch size and very high success probability? Also, are there other situations where sketches can be applied faster than simulating the execution of a turnstile streaming algorithm?

We will show that such sketches are possible in a variety of situations. We will start by addressing the fundamental heavy-hitters problem, where we give a sketch that can be applied quickly, and for which the heavy hitters can be recovered quickly. We use these algorithms as a stepping stone towards $\ell_p$ estimation for $p \geq 2$, as well as for solving least-squares regression and low-rank approximation. As a consequence of our heavy-hitters algorithm, we also observe that one can perform $\ell_p$ sampling with similar guarantees.

## 2 PRELIMINARIES

We first recall the basic setup for the problem types that we consider, as well as some of the standard results that we will apply.

$\ell_p$ **estimation.** For the $\ell_p$ estimation problem (also referred to as the $F_p$-moment estimation problem in the streaming literature), the goal is to construct a sketch $S$ that can be applied to a vector $x \in \mathbb{R}^d$, from which the $\ell_p$-norm of $x$ can be approximated.

**Problem 2.1.** (($\epsilon, \delta, \ell_p$)-norm estimation) Construct a sketch $S$, from which observing $Sx$ allows one to recover $\|x\|_p$ to within $1 \pm \epsilon$ multiplicative error. That is, with probability at least $1 - \delta$, a recovery algorithm should output an $M$ with $(1 - \epsilon) \|x\|_p \leq M \leq (1 + \epsilon) \|x\|_p$.

We will be interested in optimizing the runtime of applying the sketch $S$. We consider this problem for $p = 2$, $p \in (1, 2)$ and $p > 2$.

**Heavy-hitters** The $\ell_2$-heavy-hitters problems seeks to identify the outlying coordinates of a vector $x$. Again, we are interested in sketches that can be applied to solve this problem quickly.

**Problem 2.2.** (($\epsilon, \delta, \ell_p$)-heavy-hitters) Construct a sketch $S$ so that observing $Sx$ allows one to recover recover a set of indices $\mathcal{H}$ such that $\mathcal{H}$ contains all $i$ with $x_i \geq \epsilon \|x\|_p$. Moreover $\mathcal{H}$ should contain at most $O(1/\epsilon^p)$ elements, and the recovery should succeed with probability at least $1 - \delta$.

**Johnson-Lindenstrauss** An $(\epsilon, \delta)$ Johnson-Lindenstrauss (JL) embedding $S : \mathbb{R}^d \to \mathbb{R}^m$ preserves the $\ell_2$ norm of a given vector $x \in \mathbb{R}^d$ to within $(1 \pm \epsilon)$ distortion with failure probability at most

---

[1]under the assumption that the sketch is non-adaptive, and in the so-called "cell probe model" (see Larsen et al. (2015))

$\delta$. In other words, with failure probability at most $\delta$, $(1 - \epsilon) \|x\|_2 \leq \|Sx\|_2 \leq (1 + \epsilon) \|x\|_2$. Many Johnson-Lindenstrauss constructions are known. For example, a matrix of i.i.d. random signs allows for the optimal embedding dimension of $\frac{1}{\epsilon^2} \log \frac{1}{\delta}$ Freksen (2021).

**CountSketch**   A CountSketch as introduced by Charikar et al. (2002), works by hashing the entries of a vector $x \in \mathbb{R}^d$ into buckets, where each entry receives a random sign. Specifically we choose a hash function family $h : [d] \to [m]$ which is 2-universal (i.e., pairwise independent with $h(i)$ uniform for all $i$). We also choose random signs $\sigma_i$ which are generally taken to be 4-wise independent. Then the value of the CountSketch $Sx \in \mathbb{R}^m$ is given by $(Sx)_i = \sum_{j:h(j)=i} \sigma_j x_j$. It is well-known that a CountSketch with $m = O(\frac{1}{\epsilon^2 \delta})$ preserves the $\ell_2$ norm up to $(1 \pm \epsilon)$ distortion, with failure probability at most $\delta$ Freksen (2021). Importantly for us, CountSketch can be applied in $O(\mathrm{nnz}(x))$ time[2].

**Tensors.**   We will use tensors a few times below. If $v_1, \ldots, v_q$ are in $\mathbb{R}^d$, then we use $v_1 \otimes \cdots \otimes v_q \in \mathbb{R}^{\otimes q}$ to denote their tensor product. We use the same notation $A \otimes B$ to represent the Kronecker product of matrices. Importantly we have the identity $(A \otimes B)(v \otimes w) = (Av \otimes Bw)$. We index into a $q$ mode tensor with a multi-index $i = (i_1, \ldots, i_q)$. For the rank one tensor $v = v_1 \otimes \cdots \otimes v_q$ we have $v_i = (v_1)_{i_1} \ldots (v_q)_{i_q}$ and this definition extends linearly to tensors of higher rank.

**Regression.**   Given a matrix $A \in \mathbb{R}^{n \times d}$ and a vector $b \in \mathbb{R}^n$ the approximate regression problem asks for a vector $x$ satisfying $\|Ax - b\|_2^2 \leq (1 + \epsilon) \|Ax_* - b\|_2^2$, where $x_*$ is the least-squares solution. We will be interested in designing sketches $S$ that can can applied to $A$ and $b$ on the left, and admit recovery of an approximate regression solution $x$. Note that we are typically interested in the case when $n$ is much larger than $d$.

**Low-rank approximation.**   Given $A \in \mathbb{R}^{d \times n}$, the goal of the rank-$k$ approximation problem is to output an orthogonal projection $\Pi \in \mathbb{R}^{d \times d}$ of rank at most $k$ satisfying $\|\Pi A - A\|_F^2 \leq (1 + \epsilon) \|A - A_k\|_F^2$, where $A_k$ is the optimal rank $k$ approximation of $A$. Here we are interested in sketches that can be applied to $A$ on the left, and from which a low-rank approximation can be recovered. We are typically interested in the situation where $n$ is larger than $d$.

**Communication model.**   The main motivation for our sketches comes from distributed computation. While a variety of models could be considered, our work is primarily suited to one-way communication in the coordinator model. In this model, we imagine a collection of $s$ servers, each holding a vector (or matrix) $X_1, \ldots, X_s$. The coordinator's goal is to compute some quantity that depends on $\sum X_i$. For example, the $X_i$'s could be vectors, with the goal to approximate $\|\sum X_i\|_p$. In the one-way model, each of the servers performs some computation, and sends the result of that computation to the coordinator, who must then output a result. This is closely connected to sketching. Given a linear sketch $S$ which is drawn from some distribution of sketching matrices, the servers can compute $SX_i$, and send them to the coordinator who then computes $S \sum X_i$. Note that we assume shared randomness throughout, so that the servers can agree on a sketching matrix.

## 3   OUR RESULTS

We introduce a framework for boosting sketch performance, with our detailed results as follows.

### 3.1   HEAVY-HITTERS AND APPLICATIONS

For solving the $\ell_2$ heavy-hitters problem, we show the following:

**Theorem 3.1.** *For $\epsilon \geq d^{-0.5+c}$, there is an $\ell_2$ heavy hitters sketch with sketching dimension $O(\log d / \epsilon^2)$ that can be applied to $x \in \mathbb{R}^d$ in $O(d)$ time and such that recovery succeeds with at least $1 - d^{-c_2}$ probability.*

---

[2]We work in word-RAM model where arithmetic operations on words of size $\log(d)$ bits can be carried out in constant time. This allows hash functions to be evaluated in linear time, if implement CountSketch using a linear hash function family for example.

One drawback of this sketch is that it has slow decoding time, but by combining with Larsen et al. (2019) we can obtain both fast sketching and fast decoding time. We note that the overall combination is non-trivial, and our solution involves using appropriate hash functions for solving systems of linear equations, as detailed below.

We point out that this immediately gives an $O(d)$ time sketch that can be applied for solving the distributed heavy-hitters problem. One simply applies the sketch on each server, and sends them to the coordinator who can then aggregate and decode. This distributed version of the heavy-hitters problem has been considered before Cormode et al. (2011); Woodruff & Zhang (2012); Huang et al. (2012); however our novelty is in allowing for truly linear-time sketching by the servers, while retaining $O(1/\text{poly}d)$ failure probability as well as the fast decoding time of ExpanderSketch. We also show how to apply these ideas to recover heavy hitters via tensor sketching with low failure probability.

By applying similar ideas, but combining with a tensor sketch, we also show to recover the heavy-hitters from a tensor, via a sketch that can be applied to rank one tensors in linear time.

**Theorem 3.2.** *For $\epsilon \geq d^{-0.25}$ with $c > 0$ a constant, there is an $\epsilon$ distortion $\ell_2$ estimation sketch with sketching dimension $O(\frac{1}{\epsilon^2} \log d)$, that can be applied to a vector $x_1 \otimes \cdots \otimes x_q \in \mathbb{R}^{d^{\otimes q}}$ in $O(dq)$ time. Moreover the failure probability can be set to be $1/\text{poly}(d)$ for an arbitrary fixed polynomial.*

*Additionally, the same sketch allows for point queries. Given a multi-index $i_1, ..., i_q$ of $v$, we may output an estimate of $v_{i_1, ... i_q}$ accurate to $\epsilon \|v\|$ additive error.*

Finally, since $\ell_2$ sampling may be carried out by using heavy-hitters as a black box, we show in the appendix that an $\ell_2$ sampling sketch can be applied in our framework.

### 3.2 $F_p$ MOMENT ESTIMATION FOR $p > 2$

For $p \geq 2$, we apply our heavy-hitters sketch, along with known $F_p$-moment estimation sketches, to construct an $F_p$ moment estimation sketch that can be applied in linear time, and that succeeds with high probability:

**Theorem 3.3.** *For $p \geq 2$, there is an $F_p$ estimation sketch that can applied in time $O(d)$, and that succeeds with probability at least $1 - d^{-c}$ for an arbitrary constant $c$. Moreover, the sketch uses space $O(d^{1-2/p}\text{poly}(\log(d)/\epsilon))$ provided that $\epsilon^{1+2/p} > d^{-1/p+c_2}$.*

*As a consequence, there is a one-way communication scheme for $F_p$ moment estimation in the coordinator model that uses $O(d^{1-2/p}\text{poly}(\log(d)/\epsilon))$ communication per server, only $O(d)$ processing time by each server, and fails with probability at most $d^{-c}$ for an arbitrary constant $c$.*

We note that this sketch retains the $\tilde{O}(d^{1-2/p})$ dependence on dimension, which is tight up to log factors. While the space can be improved in by $\text{poly}(\log d/\epsilon)$ factors Ganguly (2011), as far as we know, our result that $(1 + \epsilon)$-approximate $F_p$ moment sketches can be applied in truly linear time and achieve failure probability $1/\text{poly}(d)$ is the first of its kind.

Much prior work on distributed $\ell_p$ estimation has focused on non-negative vectors (see for example Esfandiari et al. (2024) and references therein). Since we construct linear sketches, our results can all applied in the distributed setting without a non-negativity assumption.

### 3.3 LINEAR ALGEBRAIC PROBLEMS

Using similar ideas to $\ell_2$ estimation, we give a way of boosting sketches for linear regression and low-rank approximation, so that they can be applied in linear time, with arbitrary $1/\text{poly}(d)$ failure probability.

**Theorem 3.4.** *There is a sketch that gives a $(1 \pm \epsilon)$ approximate solution for least squares regression with probability at least $1 - \frac{1}{d^{c_1}}$ and with sketching dimension $O(\frac{d}{\epsilon} \log d)$, that can be applied to a matrix $A \in \mathbb{R}^{n \times d}$ in $O(\text{nnz}(A))$ time, provided that we have $\text{nnz}(A) \geq \frac{1}{\epsilon}d^{2+c_2}$. Here, $c_1, c_2 > 0$ can be taken to be arbitrary constants.*

**Theorem 3.5.** *Let $A \in \mathbb{R}^{d \times n}$ with $n \gg d$. There is an (oblivious) sketching matrix $S$ such that given $AS$ one can recover a rank $k$ orthogonal projection $\Pi$ such that $\|\Pi A - A\|_F^2 \leq (1 + \epsilon) \|A_k - A\|_F^2$, where $A_k$ is the optimal rank $k$ approximation to $A$. Moreover $S$ can be applied in $\text{nnz}(A)$ as long as $\text{nnz}(A) \geq \frac{k^2}{\epsilon^2}d^{1+c_1}$, $S$ succeeds with probability $1 - d^c$ and $S$ has sketching dimension $O(\frac{kd}{\epsilon^2} \log d)$.*

We note that the truly linear-time sketching only applies for certain parameter regimes – when $A$ is highly overdetermined for regression, and when $A$ is very wide for low-rank approximation. This is not too restrictive, however. For example, when sketching regression problems, one is typically interested in the highly over-determined case.

## 3.4 RESHAPING APPLICATIONS

In the appendix, we also show how to use a reshaping trick along with a fast matrix multiplication algorithm due to Coppersmith Coppersmith (1982) to speed up the application of an $\ell_p$ estimation sketch.

**Theorem 3.6.** *For any $1 < p \leq 2$, there is an $\ell_p$ norm estimation sketch down to dimension $O(\log \frac{1}{\delta}/\epsilon^2)$ that can be applied to $x \in \mathbb{R}^d$ in time $O\left(d(poly\log\frac{\log d}{\epsilon} + poly\log\log\frac{1}{\delta})\right)$,*

*as long as $n \geq \left(\frac{\log d \log \frac{1}{\delta}}{\epsilon}\right)^C$, for an absolute constant $C$. This sketch gives a $(1 \pm \epsilon)$-approximation with probability at least $1 - \delta$.*

While previous work has given fast update times for $\ell_p$ estimation sketches Kacham et al. (2023); Kane et al. (2011), as far as we are aware, this dependence on the failure probability within the runtime is new.

**Communication protocols.** A natural application of our results is in a communication setting, where each of several worker serves communicates one-way with a central coordinator who then aggregates their data. In this setting, one might be interested in minimizing the amount of data that the servers must transmit, as well as minimizing the amount of computation done by each server. Our results show that for some natural problems, we can design protocols where the servers can perform truly linear amounts of computation, without asymptotically increasing the communication cost. Given a linear sketch our protocol is always to perform the sketch on each server, and send the sketch to the coordinates, who then uses linearity to aggregate them. For brevity, we often do not state the communication result explicitly, but such a protocol follows from each of our results.

## 3.5 ADDITIONAL RELATED WORK

Ivkin et al. (2019) previously considered using a heavy-hitters sketch to compress gradients, lower the communication cost of SGD. This type of application is well-suited to our results – indeed their theory requires a sketch with lower failure probability. Moreover the partial are available on each server (not just via a stream), and so our sketches can be carried out in such a context. Another such scenario is the kernel classification setting studied by Mahankali & Woodruff (2021), where in the polynomial kernel setting one sees tensor-structured vectors, which can each be sketched using our techniques.

For the heavy-hitters problem, we build off of the ExpanderSketch of Larsen et al. (2019), which was the first to give nearly optimal query time. However, the heavy hitters problem has a long history, going back to Charikar et al. (2002).

Several prior works have also used related techniques when studying the distributed functional monitoring problem, where the goal is to maintain an estimate of some distributed statistic at all times Huang et al. (2012); Cormode et al. (2011); Woodruff & Zhang (2012). Note that this includes the usual communication setting, in which the statistic of interest must be estimated only once. In particular, Woodruff & Zhang (2012) studies the distributed heavy hitters problem in the functional monitoring setting.

For the heavy-hitters problem, we note that the famous Misra-Gries heavy-hitters algorithm Misra & Gries (1982) provides an alternative to our approach, but only when the vectors involved are guaranteed to be non-negative. Importantly, we allow for negative entries, and so other techniques are required.

Speeding up sketches by matrix multiplication has previously been done by Alman & Yu (2020) to obtain fast update time, by batching updates. Our matrix-multiplication techniques follow a similar principle, and are in fact somewhat simpler because we have full access to the entire vector that we

wish to sketch. In contrast Alman & Yu (2020) only sees updates over the stream, but chooses to batch updates in order to allow for a matrix multiplication speedup.

Finally, we remark that Kacham et al. (2023) gives another way to slightly speed up the sketch of Kane et al. (2011), by using their pseudorandom generator to improve the update time by a $\log \log \frac{1}{\epsilon}$ factor.

## 4 HEAVY-HITTERS AND APPLICATIONS

**Warm-up: $\ell_2$-estimation.** Our main goal of this section is a heavy-hitters sketch that can be applied in linear time, and that admits fast recovery of the heavy-hitters. We first describe a simple algorithm for heavy-hitters that allows for fast sketching, but not fast recovery. Our main novelty of this section is to show how to combine this simple heavy-hitters approach with ExpanderSketch, so that our sketch admits fast recovery.

We first describe a simple algorithm that admits a fast sketch for $\ell_2$ estimation. Our main application of this sketch is recovery of the heavy-hitters. On its own, this sketch allows fast sketching, but not fast recovery. Our main heavy-hitters result will show that we can obtain fast recovery as well by combining ExpanderSketch with our procedure, and choosing the relevant hash functions carefully.

Let $C$ be a parameter to be chosen later, and let $\delta_1$ and $\epsilon_1$ be the failure probability and distortion parameters of a Count Sketch. Also let $\delta_2$ and $\epsilon_2$ be the parameters for a dense JL sketch. Assume that $\epsilon_1$ and $\epsilon_2$ are at most 1, and that we are interested in sketching a vector $x \in \mathbb{R}^d$.

A simple observation is that a CountSketch can first be applied to $x$ to slightly reduce the dimension slightly so that the dense JL sketch can be applied in $O(d)$ time.

Take $S_1, \ldots, S_C$ to be independent Count Sketches each with $O(\epsilon_1^{-2} \delta_1^{-1})$ rows. Also take $T_1, \ldots, T_C$ to be independent JL matrices with $O(\epsilon_2^{-2} \log(1/\delta_2))$ rows.

For a fixed $x$ of appropriate dimension, consider $T_i S_i x$. By combining the guarantees for $S_i$ and $T_i$, we have $\|T_i S_i x\| = (1 \pm 2(\epsilon_1 + \epsilon_2)) \|x\|$ with failure probability at most $\delta_1 + \delta_2$. Now let $\alpha$ be the median of $\|T_1 S_1 x\|, \ldots, \|T_C S_C x\|$. The probability that the above guarantee fails on at least half the sketches is at most

$$\binom{C}{C/2} (\delta_1 + \delta_2)^{C/2} \leq (16(\delta_1 + \delta_2))^{C/2},$$

so $\alpha = (1 \pm 2(\epsilon_1 + \epsilon_2))$ with failure probability at most $(16(\delta_1 + \delta_2))^{C/2}$.

Each CountSketch $S_i$ takes $O(\mathrm{nnz}(x))$ time to apply. Each $T_i$ is sketching a vector of dimension $O(\epsilon_1^{-2} \delta_1^{-1})$ and hence can be applied in time $O(\epsilon_1^{-2} \delta_1^{-1} \epsilon_2^{-2} \log(1/\delta_2))$. In fact by using a fast JL sketch, it can be applied in time $O(\epsilon_1^{-2} \delta_1^{-1} \log(\epsilon_1^{-2} \delta_1^{-1}))$ so the total time to apply the $C$ sketches is $O(Cd + C\epsilon_1^{-2} \delta_1^{-1} \log(\epsilon_1^{-2} \delta_1^{-1}))$.

Taking $\epsilon_1 = \epsilon_2 = \epsilon \geq d^{-0.2}$, $\delta_1 = \delta_2 = d^{-(0.5-c)}$ for $c > 0$ a constant, and $C = \Theta(p)$ gives an $(O(\epsilon), d^{-p})$-$\ell_2$ estimation sketch with sketching dimension $O(p\epsilon^{-2} \log \frac{1}{\delta})$, which can be applied in $O(pd)$ time.

In the course of the above discussion, we have proven the following result.

**Theorem 4.1.** *For $\epsilon \geq d^{-0.5+c}$ with $c > 0$ a constant, there is an $\epsilon$ distortion $\ell_2$ estimation sketch with sketching dimension $O(\frac{1}{\epsilon^2} \log d)$, that can be applied to a vector $x \in \mathbb{R}^d$ in $O(d)$ time. Moreover the failure probability can be set to be $1/poly(d)$ for an arbitrary fixed polynomial.*

*More generally, we can achieve an $\epsilon$ distortion sketch with failure probability $\delta$ using a sketch of dimension $O(\frac{1}{\epsilon^2} \log \frac{1}{\delta})$ and time $O(C \, \mathrm{nnz}(x) + \epsilon^{-2} \delta^{-1/C} \log \frac{1}{\delta})$, where $C$ is a parameter that can be chosen.*

**Heavy Hitters.** By a standard argument, this approach immediately yields a heavy-hitters sketch in linear time.

**Theorem 4.2.** *For $\epsilon \geq d^{-0.5+c}$, there is an $\ell_2$ heavy hitters sketch with sketching dimension $O(\log d/\epsilon^2)$ that can be applied to $x \in \mathbb{R}^d$ in $O(d)$ time and such that recovery succeeds with at least $1 - d^{-c_2}$ probability.*

*Proof.* We first apply the $\ell_2$ estimation sketch $S$ from above with $\text{poly}(1/d)$ failure probability. Using this sketch allows us to estimate the norms $x$ and $x - e_i$ to $1 \pm \epsilon$ multiplicative error for each standard basis vector $e_i$. We have the relation

$$\langle x, e_i \rangle = \frac{1}{2}(\|x + e_i\|_2^2 - \|x\|_2^2 - \|e_i\|_2^2).$$

Each norm in this expression can be approximated to within $(1 \pm \epsilon)$ multiplicative error. To approximate $\|x + e_i\|_2$, note that we first compute $Sx + Se_i$ which we can do since $S$ and $Sx$ are both known. All of these estimates are correct with probability $O(d/\text{poly}(d)) = O(1/\text{poly}(d))$ by taking a union bound over coordinates.

This directly gives an additive $\epsilon(\|x\|^2 + 1)$ additive approximation to each entry of $x$. To obtain an $\epsilon \|x\|_2$ approximation, we would like to first normalize $x$, which we can do via our $\ell_2$ approximation sketch. In parallel, we apply the $\ell_2$ sketch to compute $Z$ with $Z^2 = (1 \pm \frac{1}{2}) \|x\|_2^2$. Now we plug $\frac{1}{Z}x$ into argument of the above paragraph, obtaining an additive $\epsilon(\frac{1}{Z^2} \|x\|_2^2 + 1) = O(\epsilon)$ approximation to each entry of $\frac{1}{Z}x$. Rescaling by $Z$ and adjusting $\epsilon$ by a constant factor yields an $\epsilon \|x\|_2$ additive approximation to each entry of $x$.

By replacing $\epsilon$ with $\epsilon/4$, this is sufficient to solve the heavy-hitters problem – one simply returns the $i$'s such that $x_i$ is estimated to be large than $\epsilon/2$. Note that there can be at most $O(1/\epsilon^2)$ such values if the estimates for all $i$ are correct. $\square$

**Corollary 4.3.** *For $p \geq 2$ and $\epsilon \geq d^{-1/p+c}$, there is an $\ell_p$ heavy hitters sketch with sketching dimension $O(\frac{1}{\epsilon^2} d^{1-2/p} \log d)$ that can be applied to $x \in \mathbb{R}^d$ in $O(d)$ time and such that recovery succeeds with at least $1 - d_2^c$ probability. Moreover this sketch recovers all coordinates to additive error $\epsilon \|v\|_p$.*

*Proof.* Recall the bound $\|v\|_p \geq d^{\frac{1}{p} - \frac{1}{2}} \|v\|_2$. This implies that an $(\epsilon, \ell_p)$-heavy-hitter is an $(\epsilon d^{\frac{1}{p} - \frac{1}{2}}, \ell_2)$-heavy-hitter. So the claim immediately follows from the $\ell_2$ heavy-hitters. $\square$

We next show how one can combine with ExpanderSketch to get fast recovery.

**Theorem 4.4.** *For $\epsilon \geq d^{-0.2}$ there is an $(\epsilon, \delta, \ell_2)$-heavy-hitters sketch that can applied to a vector $v \in \mathbb{R}^d$ in $O(d)$ time, uses $O(\frac{1}{\epsilon^2} \log d)$ space has failure probability at most $1/\text{poly}(d)$ (where $\text{poly}(d)$ can be $d^c$ for any constant $c$). The time to decode the sketch is $O(\text{poly}(\log d/\epsilon))$. More specifically, to obtain $O(d^{-p})$ failure probability, the time to decode is $O(\text{poly}(\log d)(1/\epsilon)^{O(p)})$.*

*Proof.* See appendix. $\square$

While we defer the full proof, we discuss the main idea here. As in the case of $\ell_2$ estimation, the main idea is to first apply a CountSketch to reduce the dimension slightly. The observation is that a CountSketch preserves heavy hitter coordinates with good probability. In fact, by setting the sketching dimension to be $d^{0.8}$ for example, we can ensure that each individual heavy hitter coordinate is preserved with *some* $1/\text{poly}(d)$ failure probability. Now applying the ExpanderSketch of Larsen et al. (2019) will recover these heavy elements with high probability. Note that, as with $\ell_2$ estimation, the CountSketch reduces the dimension slightly below $d$ in linear time, so that the extra overhead from applying the ExpanderSketch is $o(d)$.

As stated, this scheme only succeeds with a fixed $1/\text{poly}(d)$ failure probability. However by running this scheme multiple times in parallel, we ensure that all heavy hitters are found with arbitrarily small $1/\text{poly}(d)$ failure probability.

There is another problem when it comes to finding the heavy hitters though. The issue is that we are applying the ExpanderSketch to the CountSketch, so we are able to recover the heavy indices from the CountSketch, but we still need to recover the heavy indices from our original vector $x$. The idea is that heavy coordinates of $x$ map into heavy buckets of the CountSketch. So if we can identify the heavy buckets, then we obtain a collection of "candidates" from $x$ – namely all the entries of $x$ that hash into one of those buckets. If we only ran the scheme once, then the number of candidates would be much larger than $1/\epsilon^2$ which would be undesirable. However, as we are repeating the scheme a

large constant number of times, we can take the elements of $x$ that show up as candidates, at least half the time say. For an arbitrary hash function, this would be inconvenient as we would have to produce a list of $\text{poly}(d)$ candidates for each heavy hitter. So we instead choose a hash function with linear structure so that the hash functions are easy to invert. Given such a hash function, we can then apply a brute-force approach of looping over all subsets of half the CountSketches, and identifying coordinates of $x$ that are candidates for each sketch. Since each CountSketch only has $\text{poly}(1/\epsilon)$ heavy coordinates, we only need to loop over $\binom{C}{C/2}\text{poly}(1/\epsilon)^C$ subsets of coordinates, for a constant $C$. Identifying common candidates then amounts to inverting a linear system (over $\mathbb{F}_2$), which can be carried out efficiently.

**Extension to Tensors.** The same argument as above gives a JL sketch for tensors. To apply this, we just need the fact that a tensor product of CountSketch preserves norms. This is standard.

**Proposition 4.5.** *Let $S_1 \otimes \cdots \otimes S_q$ be a tensor product of $q$ CountSketches, each down to dimension $\frac{q}{\epsilon^2 \delta}$. For $v \in \mathbb{R}^{d^{\otimes q}}$, we have $\|S_1 \otimes \cdots \otimes S_q v\|_2^2 = (1 \pm \epsilon) \|v\|_2^2$ with probability at least $1 - \delta$.*

*Proof.* Each of our CountSketches has the $(\epsilon, \delta, 2)$-JL-moment property. (See Woodruff et al. (2014) for example). Additionally Ahle et al. (2020) shows that the tensor product of $q$ sketches each with the $(\epsilon, \delta, 2)$-JL moment property has the $(\sqrt{q}\epsilon, \delta, 2)$-JL moment property. So it is sufficient for each of our sketches to have the $(\epsilon/\sqrt{q}, \delta, 2)$-JL moment property. $\square$

Now the same argument from above applies to give the following.

**Theorem 4.6.** *For $\epsilon \geq d^{-0.25}$ with $c > 0$ a constant, there is an $\epsilon$ distortion $\ell_2$ estimation sketch with sketching dimension $O(\frac{1}{\epsilon^2} \log d)$, that can be applied to a vector $x_1 \otimes \cdots \otimes x_q \in \mathbb{R}^{d^{\otimes q}}$ in $O(dq)$ time. Moreover the failure probability can be set to be $1/\text{poly}(d)$ for an arbitrary fixed polynomial.*

*Additionally, the same sketch allows for point queries. Given a multi-index $i_1, ..., i_q$ of $v$, we may output an estimate of $v_{i_1, ... i_q}$ accurate to $\epsilon \|v\|$ additive error.*

*Proof.* As above, sample $C$ independent tensor products of CountSketch $Q_1, \ldots, Q_C$, with $\delta$ set to $d^{-0.25}$. Each of these sketches can be applied in $O(nq)$, and the sketching dimension is $d^{0.25}q/\epsilon^2$. Now the resulting sketch can be composed with a TensorSketch of Ahle et al. (2020), which now takes $o(dq)$ time.

The failure probability is $d^{-0.25}$, so repeating $C$ times and taking the median boosts the failure probability to $1/d^c$ for any choice of $c$. (Note that $C$ depends on c.)

Finally, exactly the same argument that we used above for heavy-hitters shows that our sketch supports point queries. $\square$

The same ideas also allow us to construct fast tensor sketches with low failure probability and fast recovery.

**Theorem 4.7.** *There is a sketch that solves the $\ell_2$-heavy-hitters problem with $O(q/\epsilon^2 \log(d))$ space and failure probability $d^{-c}$ for any fixed constant c. Moreover this sketch can be applied to rank one tensors in linear time, and the heavy-hitters can be decoded in $O(\epsilon^{-2q})$ time.*

### 4.1 APPLICATION TO COMMUNICATION

One can apply this result in a distributed setting. Suppose that server could form their sketch in linear time, send it to the coordinator using $O((1/\epsilon^2) \log d)$ communication, who can then merge the sketches and decode in $\text{poly}(\log d/\epsilon)$ time, with failure probability $1/d^3$ say. Forming the ExpanderSketches directly would have required $d \log d$ time on the servers' sides, but with our approach, linear time is possible.

**Corollary 4.8.** *Suppose that Alice and Bob hold vectors $v_A$ and $v_B$ in $\mathbb{R}^d$. Then there is a communication protocol that allows Alice to output a set $\mathcal{H}$ of indices, such that $|\mathcal{H}|$ is $O(1/\epsilon^2)$.*

*The protocol is one-way with Bob sending his message to Alice, and uses $O(\log d/\epsilon^2)$ words[3] of communication. Moreover the total time required for Bob to compute his message is $O(d)$, and the time required for Alice to generate $\mathcal{H}$ is $O(poly(\log d/\epsilon))$.*

*Proof.* Bob simply computes the heavy-hitters sketch discussed above, and sends the sketched vector $Sv_B$ to Alice, who can then compute $S(v_A + v_B)$, and decode as above. □

We remark that one could equally well consider an $s$-player communication game, where each player communicates with a central coordinator. The protocol is precisely the same. We just state the two-player version for simplicity.

## 5 $F_p$ MOMENT ESTIMATION.

In this section, we construct fast sketches for $F_p$ estimation. The following lemma is simply an application of Bernstein's inequality, which shows that the $F_p$ moment of a flat vector is approximately preserved with high probability, when we subsample at a high enough rate.

**Lemma 5.1.** *Let $v \in \mathbb{R}^d$ be an arbitrary vector. Let $S$ be a random subset of $[d]$ where each coordinate is included independently with probability $q$. Also suppose that $\|v\|_\infty \leq M$. Then with probability at least $\delta$ we have $\frac{1}{q} \|v_S\|_p^p = (1 \pm \epsilon) \|v\|_p^p$ provided that $M^p \leq \frac{q\epsilon^2}{4 \log \frac{1}{\delta}} \|v\|_p^p$.*

*Proof.* See appendix □

Using this, we show the following result for $F_p$ moment estimation.

**Theorem 5.2.** *For $p \geq 2$, there is an $F_p$ estimation sketch that can applied in time $O(d)$, and that succeeds with probability at least $1 - d^{-c}$ for an arbitrary constant $c$. Moreover, the sketch uses space $O(d^{1-2/p}poly(\log(d)/\epsilon))$ provided that $\epsilon^{1+2/p} > d^{-1/p+c_2}$.*

*As a consequence, there is a one-way communication scheme for $F_p$ moment estimation in the coordinator model that uses $O(d^{1-2/p}poly(\log(d)/\epsilon))$ communication per server, only $O(d)$ processing time by each server, and fails with probability at most $d^{-c}$ for an arbitrary constant $c$.*

We defer the proof to the appendix, but describe the approach here. The typical approach for $F_p$ moment estimation of $v \in \mathbb{R}^d$, introduced by Indyk & Woodruff (2005), is to partition the coordinates of $v$ into approximate level sets, and then to approximate the size of each level set. Estimating the level set sizes can be done by subsampling at geometric rates, and then extracting the $\ell_p$ heavy-hitters for each sample. This gives a sketch that can be applied in $O(dpoly(\log d/\epsilon))$ time, with $9/10$ success probability. Our goal is to boost this to $d^{-c}$ failure probability, and linear sketching time without increasing the space too much.

The key intuition is that the highest sampling levels contribute the bulk of the runtime of $F_p$ estimation sketches. These top sampling levels are needed to catch any heavy-hitters of $v$. However, we can recover the heavy-hitters directly, using our $\ell_p$-heavy-hitters sketch which achieves high success probability and linear time. With the heavy-hitters gone, the $F_p$ estimation sketch on the remaining coordinates of $v$ can be carried out in say $O(d/\log d)$ time. Thus we repeat this sketch $O(\log d)$ times to boost to $1/poly(d)$ failure probability, without exceeding linear time overall

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

## A    ADDITIONAL TECHNICAL OVERVIEW

We outline several main ideas behind our results.

$\ell_2$**-heavy-hitters**    We obtain our $\ell_2$-heavy-hitters sketch as a consequence of norm estimation. One approach to sketch the $\ell_2$ norm for a vector $v \in \mathbb{R}^d$ with $1/\mathrm{poly}(d)$ failure probability is to apply a dense JL sketch – for example a matrix $S \in \mathbb{R}^{m \times d}$ with i.i.d. random signs where $m = O(\frac{1}{\epsilon^2} \log d)$. However this matrix-vector product takes $O(d \log d / \epsilon^2)$ time. This can be improved by choosing a structured matrix for the JL embedding. For example, if one chooses a random-sign Toeplitz matrix, with $O(\frac{\log^2 d}{\epsilon^2})$ rows (which is known to be a JL embedding Freksen & Larsen (2020)), then the matrix-vector multiplication can be carried out in $O(d \log d)$ time Kailath & Sayed (1999). [4]

We aim to give an $\ell_2$ estimation sketch which runs in $O(d)$ time, with no additional logarithmic factors. Our approach is simple, but appears to be new and is a starting point for our later results.

---

[4] In fact, one can do slightly better. By decomposing $S$ into $d/m$ blocks of size $m \times m$, each of which is Toeplitz, multiplication by $S$ can be carried out in $O((d/m)m \log m) = O(d \log m)$ time Freksen (2021). For $m = O(\log d/\epsilon^2)$, this would give a runtime of $O(d(\log \log d + \log \frac{1}{\epsilon}))$.

Roughly, we would like to first apply a CountSketch to reduce the dimension of $v$ slightly. The issue is a CountSketch with distortion $\epsilon$ and failure probability $\delta$, needs $O(\frac{1}{\epsilon^2 \delta})$ rows. This means that it is not possible to achieve arbitrary $1/\mathrm{poly}(d)$ failure probability initially. For instance, to achieve a failure probability of $1/d^2$, we would need to take $m = d^2$, which is useless. However it is easy to achieve *some* $1/\mathrm{poly}(d)$ failure probability from a CountSketch, while getting some dimension reduction. For example, if we are willing to sketch down to dimension $n^{0.8}$, then we could achieve, say $\delta = n^{-0.5}$ failure probability, as long as $1/\epsilon$ is not too large. Such a sketching dimension is unacceptably large, but now that the dimension has been reduced slightly, we are free to compose with a dense JL sketch, and pay no additional asymptotic cost (since in our example $n^{0.8} \log n = o(n)$). To achieve arbitrary $1/\mathrm{poly}(n)$ failure probability, we can then boost this failure probability by repeating this scheme a constant number of times and use the median as our $\ell_2$ estimate.

As a consequence of having an $\ell_2$ estimation sketch, we obtain an $\ell_2$ heavy hitters sketch almost for free, using a standard approach. To compute the $\epsilon$-heavy-hitters of $v$ it suffices to be able to estimate the values of $\langle e_i, v \rangle$ to within $O(\epsilon \|v\|)$ additive error, simultaneously for all $i$. This inner product can be expressed as a sum and difference of $\ell_2$ norms, each of which can be approximated using the $\ell_2$ estimation sketch. Since each relevant $\ell_2$ estimation problem can be solved accurately with arbitrary $1/\mathrm{poly}(d)$ failure probability, the same is true for the heavy-hitters problem after union bounding over $d$ events. A black-box $\ell_2$ heavy-hitters algorithm suffices to solve $\ell_p$-hitters by standard results. So we in fact obtain an optimal linear-time, high-probability sketch for all $p \geq 1$.

**Linear regression and low-rank approximation.** Our algorithms for linear regression and low-rank approximate follow a similar template as for $\ell_2$ approximation. For approximately solving the linear regression problem $\min_x \|Ax - b\|_2$, we recall that a CountSketch with $\frac{d^2}{\epsilon \delta}$ rows is a $(1 \pm \epsilon)$ distortion subspace embedding with probability at least $1 - \delta$ by ideas of Woodruff & Zhang (2013), which is sufficient to solve regression. This sketch has the advantage of running in $\mathrm{nnz}(A)$ time, but the resulting sketching dimension is suboptimal, both in terms of $d$ and $\delta$. We therefore compose this sketch with a fast JL sketch that runs in slightly super-linear time, but achieves the optimal $O(\frac{d}{\epsilon^2})$ dimension. Since this sketch is applied to the CountSketch of $A$, which has fewer rows then $A$, we maintain linear runtime, at least when $A$ has many rows. The failure probability is dominated by the CountSketch, and we would ideally like to boost this by running the above procedure a constant $C$ number of times and choosing the best regression solution among our candidates $x_1, \ldots x_C$. In other words, we would like an $\ell_2$ estimation sketch $S$ that allows us to estimate $\|Ax_i - b\|_2^2 \approx \|S(Ax_i - b)\|_2^2$ with high probability. To accomplish this, we reuse use our $\ell_2$ estimation procedure discussed above. Then $S$ can be applied to $A$ and $b$ in parallel with the subspace embedding, and we can select the $x_i$ for which the sketched estimate for the mean-squared error is smallest.

Our approach for low-rank approximation follows a very similar template for regression. The main difference is that low-rank approximation requires a sketching primitive which is slightly stronger than a subspace embedding (we use the notion of a projection-cost preserving sketch Musco & Musco (2020), although various combinations of sketching primitives would work here). Fortunately, the composition of a CountSketch with a fast JL sketch has the required properties for low-rank approximation, and so the same approach as for regression applies. We use the same idea of applying our $\ell_2$ approximation sketch on the left to estimate the Frobenius error of candidate low-rank approximations. This allows us to (approximately) select the best solution obtained from a constant number of independent trials.

**Speeding up sketches by reshaping.** We consider the problem of computing an $\ell_p$-norm estimation sketch for a vector $x$, where $p \in (1, 2]$ with high probability. One could take $S$ to be an $\ell_p$ estimation sketch of Kane et al. (2011) for example, and then repeat $\log \frac{1}{\delta}$ times. However this would give a sketch whose application time has a multiplicative $\log \frac{1}{\delta}$ dependence. We aim for something faster.

To see the main idea, suppose that we have an embedding sketch $T : \ell_p \to \ell_r$ (note that the sketch of Kane et al. (2011) is not an embedding). Then we could compute $Tx$ directly. However, we are also free to reshape $x$ into a matrix $X$, and then adjust the dimensions of $T$ appropriately. The advantage is that we are now computing a matrix product $TX$, and so we can apply a known fast matrix multiplication algorithm. Indeed, if the ratio of columns to rows for $T$ is sufficiently large (i.e., larger than some $d^c$ for some $c$), then multiplication by $T$ may be carried out in roughly linear time

using an algorithm of Coppersmith (1982). Combining this reshaping idea with known embeddings $\ell_p \to \ell_r$ allows us to reduce the number of rows of $X$, while preserving its entrywise $\ell_p$ norm. Once we have reduced the dimensionality, the $\ell_p$ norm estimation sketch of Kane et al. (2011) now runs faster, and so we can apply it directly.

# B $F_p$ ESTIMATION PROOFS

**Proof of Lemma 5.1** .

*Proof.* Let

$$X = \sum_{i=1}^{d} \sigma_i v_i^p$$

where $\sigma_i \sim \frac{1}{q}\text{Bernoulli}(q)$. Note that $\text{E}(X) = \|v\|_p^p$. Set $X_i = \sigma_i v_i^p$, and note that

$$\text{E}(|X_i - v_i^p|^2) \le \text{E}(X_i^2) = \frac{1}{q}v_i^{2p}.$$

By Bernstein's inequality,

$$\Pr\left(\left|\|v\|_p^p - X\right| \ge t\right) \le \exp\left(-\frac{\frac{1}{2}t^2}{\sum_{i=1}^{d}\text{E}(|X_i - v_i^p|^2) + \frac{1}{3}\frac{1}{q}M^p t}\right)$$

$$\le \exp\left(-\frac{\frac{1}{2}t^2}{\frac{1}{q}\|v\|_{2p}^{2p} + \frac{1}{3}\frac{M^p}{q}t}\right)$$

$$\le \exp\left(-\frac{\frac{1}{2}t^2 q}{M^p\|v\|_p^p + \frac{1}{3}M^p t}\right)$$

Setting $t = \epsilon\|v\|_p^p$ with $\epsilon < 1$ gives

$$\Pr\left(\left|\|v\|_p^p - X\right| \ge \epsilon\|v\|_p^p\right) \le \exp\left(-\frac{1}{4}\frac{\epsilon^2\|v\|_p^{2p}q}{M^p\|v\|_p^p}\right) = \exp\left(-\frac{1}{4}\frac{\epsilon^2\|v\|_p^p q}{M^p}\right).$$

This rearranges to the stated claim. $\square$

We will use the following simple, technical fact below.

**Proposition B.1.** *Suppose that $v_1$ and $v_2$ have disjoint supports with $v_1 + v_2 = v$. Also suppose that $\left|\alpha_i - \|v_i\|_p\right| \le \epsilon\|v\|_p$ for $i = 1, 2$. Then*

$$\|v\|_p = (\alpha_1^p + \alpha_2^p)^{1/p} \pm 2^{1/p}\epsilon\|v\|_p.$$

*Proof.* First note that

$$\|v\|_p = (\|v_1\|_p^p + \|v_2\|_p^p)^{1/p} = \left\|[\|v_1\|_p, \|v_2\|_p]\right\|_p$$

This later two-dimensional vector differs coordinate-wise from $[\alpha_1, \alpha_2]$ by at most $\epsilon\|v\|_p$. Hence

$$\left|\left\|[\|v_1\|_p, \|v_2\|_p]\right\|_p - \|[\alpha_1, \alpha_2]\|_p\right| \le 2^{1/p}\epsilon\|v\|_p,$$

which implies the claim. $\square$

**Proof of Theorem 5.2.**

*Proof.* Write $v = v_H + v_L$ where $H$ is the coordinates of $v$ larger than $M$ and $L$ is the remaining coordinates of $v$. Let $v'$ be a random vector where $v'_i$ is $(v)_i$ with probability $q$ and $0$ otherwise. Define $v'_L$ and $v'_H$ similarly.

As in the previous lemma, we choose $M$ so that

$$M^p = \frac{q\epsilon^2}{4\log\frac{1}{\delta}} \|v\|_p^p := \alpha \|v\|_p^p.$$

Then by the previous lemma, we have

$$\frac{1}{q} \|v'_L\|_p^p = (1 \pm \epsilon) \|v_L\|_p^p,$$

with probability at least $1 - \delta$.

Note that each element of $v_H$ is an $(\alpha, \ell_p)$-heavy-hitter. With this decomposition, we show how to estimate the $F_p$ moments for $v_H$ and $v_L$.

**Estimating $v_H$.** Our $\ell_p$-heavy-hitters sketch (see Corollary 4.3), allows us to estimate all coordinates of $v$ to within $\beta \|v\|_p$ additive error and failure probability $1/\text{poly}(d)$ using space $O(\frac{1}{\beta^2} d^{1-2/p} \log d)$, and $O(d)$ sketching time, provided that $\beta \geq d^{-1/p+c}$.

Note that $v_H$ has at most $1/\alpha$ nonzero coordinates. Let $\hat{v}_H$ be the recovered heavy-hitter coordinates. Then

$$\|\hat{v}_H - v_H\|_p \leq \beta \|v\|_p \frac{1}{\alpha^{1/p}}.$$

Therefore, by the triangle inequality,

$$\|\hat{v}_H\|_p = \|v_H\|_p \pm \frac{\beta}{\alpha^{1/p}} \|v\|_p.$$

Setting $\beta = \epsilon\alpha^{1/p}$ then gives

$$\|\hat{v}_H\|_p = \|v_H\|_p \pm \epsilon \|v\|_p.$$

**Estimating $\|v'\|_p^p$.** To handle $v'$ we use a standard $F_p$-moment estimation sketch applied to $v'$. For concreteness, we can apply the sketch of Indyk & Woodruff (2005) which can be applied in time $T(d, \epsilon, p) := d\text{poly}(\log d/\epsilon)$ to a vector of length $d$, and succeeds with $9/10$ probability. We will apply this sketch to $v'$ which has $O(qd)$ entries with high probability, and hence the time to apply it is $T(qd, \epsilon, p)$.

This gives a $(1 \pm \epsilon)$ multiplicative approximation to $\|v'\|_p$. In the case that the support of $v'$ intersects the support of $v_H$ we may use our additive approximations to the coordinates of $v_H$ as above to approximate $\|v'_H\|_p$ to within additive $\epsilon \|v\|_p$ error. Since $v' = v'_H + v'_L$, this yields an additive $\epsilon \|v\|_p + \epsilon \|v'\|_p \leq 2\epsilon \|v\|_p$ approximation to $\|v'_L\|_p$.

Now recall that

$$\frac{1}{q} \|v'_L\|_p^p = (1 \pm \epsilon) \|v_L\|_p^p.$$

We now have an additive $\frac{2\epsilon}{q^{1/p}} \|v\|_p$ approximation to $\frac{1}{q^{1/p}} \|v'_L\|_p$, which for constant $p$ is an $O(\epsilon \|v_L\|_p)$ additive approximation to $\|v_L\|_p$ by the equation above. So we may approximation $\|v_L\|_p$ to additive error $O(\epsilon \|v_L\|_p + \frac{2\epsilon}{q^{1/p}} \|v\|_p) = O(\frac{\epsilon}{q^{1/p}} \|v\|_p)$.

Combining the bounds, we have a $(1 \pm O(\frac{\epsilon}{q^{1/p}}))$ multiplicative approximation to $\|v\|_p$ with constant probability. By running this algorithm $O(\log\frac{1}{\delta})$ times in parallel and taking the median answer, we boost the failure probability to $\delta$.

**Space and runtime.**   We set $\delta = d^c$. The total runtime for applying our sketch is

$$O(T(qd, \epsilon, p) \log d + d).$$

since we apply the heavy-hitter sketch in $O(d)$ time and then $F_p$ estimation sketches. To make this run in linear time, we set $q = O(\text{poly}(\log d, \epsilon))$

We set $\delta = d^c$. After unwinding all the variables, our total space is

$$O\left(d^{1-2/p}\text{poly}(\log d/\epsilon)) + \frac{1}{\epsilon^{2+4/p}}d^{1-2/p}(\log d)^{1+2/p}q^{-2/p}\right).$$

Plugging in our value for $q$ and replacing $\epsilon$ with $\epsilon q^{1/p}$ gives the desired bound.

$\square$

## C   SPEEDING UP SKETCHES BY RESHAPING

We first recall a result due to Coppersmith on fast rectangular matrix multiplication, which implies that one can multiply a $d \times d$ and $d \times d^\alpha$ matrix using $O(d^2 \log^2 d)$ multiplications for $\alpha \leq 0.17$ Coppersmith (1982). We use a small variant of this result by Williams (2014); Williams which states that a $d \times d$ times $d \times d^\alpha$ matrix multiplication can be carried out in $O(d^2\text{poly} \log d)$ time. The subtle difference between these results is that it could a priori be the case that the arithmetic circuit implicit in Coppersmith's algorithm takes a lot of time to construct. One would not expect this to be the case however, and indeed it is not. We state this fact precisely for use below.

**Theorem C.1.** *There is an algorithm for multiplying a $d \times d$ and a $d \times d^\alpha$ matrix in $O(d\text{poly} \log d)$ time, under the assumption that field operations can be carried out $O(1)$ time.*

We can use reshaping, along with this fact, to speed up sketches for $\ell_p$ embeddings.

We first recall a result for constructing embeddings $\ell_p \to \ell_r$ which follows from the proof Lemma 18 contained in Li et al. (2023).

**Lemma C.2.** *Suppose that $p > r > 1$, and let $r' \in (r, p)$ be arbitrary. There is a distribution over sketching matrices $T \in R^{m \times n}$ with $m = O(\log(1/\delta)/\epsilon^{C(\epsilon,r)})$, such that for any $v \in \mathbb{R}^n$, $\|Tv\|_r \geq (1-\epsilon) \|v\|_p$ with $1-\delta$ probability, and we have a moment bound of the form $\mathrm{E} \left| \|Ty\|_r^r - \|y\|_p^r \right|^{r'/r} \leq C \|y\|_p^{r'} /m^{(r'/r)-1}$, where $\mathrm{E} \|Ty\|_r^r = \|y\|_p^r$.*

Using this, we show the following result.

**Theorem C.3.** *For any $1 < p \leq 2$, there is an $\ell_p$ norm estimation sketch down to dimension $O(\log \frac{1}{\delta}/\epsilon^2)$ that can be applied to $x \in \mathbb{R}^d$ in time $O\left(d(\text{poly} \log \frac{\log d}{\epsilon} + \text{poly} \log \log \frac{1}{\delta})\right)$, as long as $n \geq \left(\frac{\log d \log \frac{1}{\delta}}{\epsilon}\right)^C$, for an absolute constant $C$. This sketch gives a $(1 \pm \epsilon)$-approximation with probability at least $1 - \delta$.*

*Proof.* Pick an arbitrary $r$ with $1 < r < p$. The first step is to construct an embedding $\ell_p \to \ell_r$ that reduces the dimension slightly and that can be applied quickly.

Let $T$ to be a randomized embedding $\ell_p \to \ell_r$. By Lemma C.2 there is such a $T$ with $m = O(\log n/\epsilon^{C(\epsilon,r)})$ rows, such that the stated guarantees in the lemma hold, and so that the lower bound on $\|Tv\|_r$ holds with failure probability at most 0.9.

We take $k$ independent copies of this sketch, $T^{(1)}, \ldots, T^{(k)}$ in parallel, so that our total sketching dimension is now $mk$. Call this matrix gotten by stacking the $T^{(i)}$'s $S$.

Next we reshape $x$ into a matrix $X$ with dimensions $(mk)^c \times (d/(mk)^c)$, where $c > 1/0.17$ so that the fast matrix multiplication algorithm discussed above can be applied. (If the dimensions of $X$ are not evenly divisible by $(mk)^c$, then we first pad with an appropriate number of zeroes, which does not affect the norm.) Note that our sketching matrix now has dimension $mk \times (mk)^c$. Note that for this step to work, we require $d \geq (mk)^{2c}$.

To compute $SX$, we can partition $X$ into $d/(mk)^{2c}$ matrices each of dimension $(mk)^c \times (mk)^c$. Multiplying each submatrix by $S$ takes $O((mk)^{2c}\text{poly}\log((mk)^{2c}))$ time by Theorem C.1, and there are $O(d/(mk)^{2c})$ such sub-matrices. So the total time to compute $SX$ is $O(d\text{poly}\log(mk))$.

Next we observe that Lemma C.2 shows that each $T^{(i)}$ gives an embedding of $X$ into $\ell_r$. To see that $X$ does not contract much, note that we can take $\delta$ to be $O(1/d)$ so that all columns $x_j$ of $X$ satisfy $\left\|T^{(i)}x_j\right\|_r \geq (1-\epsilon)\left\|x_j\right\|_p$ with 0.9 probability. Since $p \geq r$, this implies the desired lower bound.

To bound dilation of $X$, note that the moment bound for each column given in Lemma C.2, extends to give the same moment bound for $\|X\|_r^r$ up to constants (see Lemma 27 of Li et al. (2023) or von Bahr & Esseen (1965)). Combining with a Markov bound, this shows that $\|TX\|_r \leq (1+\epsilon)\|X\|_p$ with 0.9 probability.

At this point we have constructed $k$ independent embeddings of $x$ into $\ell_r$, each correct with 0.9 probability. Note that $m = O(\log n/\text{poly}(\epsilon))$, and so our runtime so far is

$$O(d\text{poly}\log(\log(d)k/\text{poly}(\epsilon)))$$

Note that the dimension of $SX$ is $(mk)(d/(mk)^c) = (mk)^{1-c}d$, and that the dimension of each $T^{(i)}X$ is $\frac{1}{k}(mk)^{1-c}n$.

Since we have $k$ embeddings into $\ell_r$, we can now compose each of them with an $\ell_r$ estimation sketch of Kane et al. (2011). This sketch has an update time of $\log^2 \frac{1}{\epsilon} \log\log \frac{1}{\epsilon}$ and hence can be applied to $T^{(i)}X$ in

$$O\left(\frac{1}{k}(mk)^{1-c}d\text{poly}\log\frac{1}{\epsilon}\right)$$

time, so the total time to sketch all of the $T^{(i)}x$'s is

$$O\left((mk)^{1-c}d\text{poly}\log\frac{1}{\epsilon}\right).$$

Finally, we have constructed a sketch down to dimension $O(k/\epsilon^2)$, and which yields a $(1 \pm \epsilon)$ multiplicative approximation to $v$ with at 0.8 least probability. By taking the median estimator over $k$ trials, this gives an overall failure probability of $1 - \delta$ by choosing some $k = O(\log \frac{1}{\delta})$.

Our overall sketching dimension was $O(\log \frac{1}{\delta}/\epsilon^2)$. The time to apply $S$ was

$$O(n\text{poly}\log(mk)) = O(d\text{poly}\log\frac{\log d \log \frac{1}{\delta}}{\epsilon}),$$

and the time to apply the $\ell_r$ estimation sketch was

$$O\left((mk)^{1-c}d\text{poly}\log\frac{1}{\epsilon}\right).$$

Note that the time to apply $S$ dominates as long as $mk > 1/\epsilon$ say (since $1 - c < -4$ by our choice of $c$), which we are free to assume.

$\square$

The rough idea is to first pick an arbitrary $r$ with $1 < r < p$. We first construct an embedding $\ell_p \to \ell_r$ that reduces the dimension slightly and that can be applied quickly. We let $T$ be a randomized embedding $\ell_p \to \ell_r$ and then take $k$ independent copies of this sketch, $T^{(1)}, \ldots, T^{(k)}$ in parallel, so that our total sketching dimension is now $mk$. Call this matrix obtained by stacking the $T^{(i)}$'s $S$. We then reshape $x$ into a matrix $X$ with dimensions $(mk)^c \times (d/(mk)^c)$, where $c > 1/0.17$ so that the fast matrix multiplication algorithm discussed above can be applied.

## D  REGRESSION AND LOW RANK APPROXIMATION

### D.1  REGRESSION

In this section, we observe that the same idea we used for speeding up heavy-hitters sketches can be used to speed up sketches for linear regression. The idea is to run several CountSketches in parallel. Each CountSketch is a subspace embedding with probability $1/n$ and

We first recall a well-known fact, essentially due to Woodruff & Zhang (2013), that CountSketch is a subspace embedding for $A \in \mathbb{R}^{n \times d}$ when $S$ has $d^2/(\epsilon\delta)$ rows.

**Lemma D.1.** *For $A \in \mathbb{R}^{n \times d}$ and $b \in \mathbb{R}^n$, and let $S$ be a CountSketch matrix with $O(\frac{d^2}{\epsilon\delta})$ rows. Then $S$ is an oblivious subspace embedding with distortion $(1 \pm \sqrt{\epsilon})$ for $d$-dimension subspaces, with probability at least $1 - \delta$.*

*Also, suppose that $\hat{x}$ minimize $\|S(Ax - b)\|_2^2$ then with probability at least $1 - \delta$,*

$$\min_x \|A\hat{x} - b\|_2^2 \leq (1 \pm \epsilon) \min_x \|Ax - b\|_2^2.$$

*Proof.* Theorem 9 of Woodruff et al. (2014) states that a CountSketch with $O(\frac{d}{\epsilon\delta})$ rows is an $O(1 \pm \epsilon)$ distortion subspace embedding for the column span of $A$ with probability at least $1 - \delta$. It is known that this $(1 \pm \sqrt{\epsilon})$ distortion subspace embedding suffices for obtaining a $(1 \pm \epsilon)$ approximate regression solution (see Bourgain et al. (2015)), giving the lemma. $\qquad\square$

By applying this lemma for a constant number of repetitions, and applying a JL sketch on each repetition, we can keep the sketching time at truly $\mathrm{nnz}(A)$ (with no log factors), while obtaining an arbitrary $1/\mathrm{poly}(d)$ failure probability, at least provided that $A$ is quite tall.

**Theorem D.2.** *There is a sketch that gives a $(1 \pm \epsilon)$ approximate solution for least squares regression with probability at least $1 - \frac{1}{d^{c_1}}$ and with sketching dimension $O(\frac{d}{\epsilon} \log d)$, that can be applied to a matrix $A \in \mathbb{R}^{n \times d}$ in $O(\mathrm{nnz}(A))$ time, provided that we have $\mathrm{nnz}(A) \geq \frac{1}{\epsilon}d^{2+c_2}$. Here, $c_1, c_2 > 0$ can be taken to be arbitrary constants.*

**Remark D.3.** *Here we state conditions under which we obtain truly linear sketching time. The proof of the theorem still gives a (more complicated) time complexity bound even when this condition does not hold.*

*Proof.* For fixed $\delta_1$ and $\epsilon$ we first apply $T$ sketching matrices $S_1, S_2, \ldots, S_T$ in parallel.

Each $S_i$ is gotten by first taking a CountSketch down to dimension $O(\frac{d^2}{\epsilon\delta_1})$ followed by fast JL sketch down to dimension $O(\frac{d}{\epsilon} \log \frac{d}{\delta_1})$. Since we sketch all columns of $A$ (as well as $b$), the final sketching dimension is $O(\frac{d^2}{\epsilon} \log \frac{d}{\delta_1})$ for each $S_i$. Each of these gives a $1 \pm \epsilon$ approximate regression solution with probability at least $1 - \delta_1$.

Let the resulting solutions to the sketched regression problem be $x_1, \ldots, x_T$, so that we have

$$\|Ax_i - b\|_2^2 \leq (1 \pm \epsilon) \min_x \|Ax - b\|_2^2,$$

with probability at least $1 - \delta_1$ for $i \in [T]$. Say that a regression solution is good if the above bound holds. Note that at least one of the regression solutions is good with probability at least $1 - \delta_1^T$. We set $\delta_1 = \delta^{1/T}$, so that at least one of the regression solutions is good with probability $1 - \delta$.

We would like to have an accurate estimate of $\|Ax_i - b\|_2^2$ for all $i$ so that we can pick the best solution. To do this, we run our $\ell_2$ estimation sketch from above in parallel with failure probability $\delta/T$ (note that we sketch $A$ on the left, as well as $b$, so our sketching dimension and runtime are scaled by $d$). This takes $O(\frac{d}{\epsilon^2} \log \frac{T}{\delta})$ space and requires only $O(C \mathrm{nnz}(A) + \frac{d}{\epsilon^2}(\delta/T)^{-1/C} \log \frac{T}{\delta})$ time, where $C$ is a parameter that we can choose.

All $\ell_2$ estimates are accurate to within $1 \pm \epsilon$ multiplicative error with probability all but $\delta$. Conditioned on these $\ell_2$ approximations all being correct, we then have $(1 \pm \epsilon)$ approximations to all $\|Ax_i - b\|_2^2$, so we may simply return the $x_i$ with the smallest such value. If at least one of the $x_i$'s is good, then this is guaranteed to give a $(1 \pm \epsilon)^2 = (1 \pm O(\epsilon))$ approximate regression solution.

Our overall failure probability comes from having one of the $\ell_2$ sketches fail, or all of the regression sketches $S_i$ fail. By a union bound, the total failure probability is at most

$$T\frac{\delta}{T} + \delta_1^T = O(\delta).$$

Our overall sketching dimension is

$$O(\frac{d}{\epsilon^2} \log \frac{T}{\delta} + \frac{Td^2}{\epsilon} \log \frac{d}{\delta_1}) = O(\frac{d}{\epsilon^2} \log \frac{T}{\delta} + \frac{Td^2}{\epsilon} \log d + \frac{d^2}{\epsilon} \log \frac{1}{\delta}).$$

Finally the time to apply[5] all the $S_i$'s is

$$O(T(\text{nnz}(A) + d\frac{d^2}{\epsilon\delta^{1/T}} \log \frac{d^2}{\epsilon\delta^{1/T}})),$$

and the time to apply the $\ell_2$ estimation sketch is

$$O(C \text{ nnz}(A) + \frac{d}{\epsilon^2}(\delta/T)^{-1/C} \log \frac{T}{\delta}).$$

For $\delta = d^{-c}$, $T$ and $C$ can be taken to be constants. For the parameter regime given in the theorem statement, this is $O(\text{nnz}(A))$ time as desired. $\qquad\square$

## D.2 LOW RANK APPROXIMATION

In this appendix section, we observe that ideas similar to those used to speed up our heavy-hitters sketch, can also be applied to linear algebraic problems. We first recall a few preliminary definitions and results that will be helpful here.

**Definition D.4.** *Let $S \in \mathbb{R}^{m \times d}$ be a sketching matrix. $S$ is said to satisfy the $(\epsilon, \delta, 2)$-JL moment property if for all unit vectors $x \in \mathbb{R}^d$,*

$$\mathrm{E}\left|\|Sx\|_2^2 - 1\right|^2 \leq \epsilon^2\delta.$$

*The following fact is standard. See Woodruff et al. (2014) for example.*

**Lemma D.5.** *The $(\epsilon, \delta, 2)$-JL-moment property is satisfied by a CountSketch with $O(\frac{1}{\epsilon^2\delta})$ rows, as well as a fast JL sketch with $O(\frac{1}{\epsilon^2} \log \frac{1}{\delta})$ rows.*

*We also recall the notion of a projection-cost-preserving (PCP) sketch Musco & Musco (2020); Cohen et al. (2015), which is well-suited to low-rank approximation problems. (One could use other combinations of sketching primitives here, but the explicit conditions given in Musco & Musco (2020) are convenient to check.) We simplify the standard definition slightly for our context. In what follows, we will think of $A$ as having a large number columns representing feature vectors for example. So $A$ should be thought of as very wide.*

**Definition D.6.** *Let $A \in \mathbb{R}^{d \times n}$. The sketching matrix $S \in \mathbb{R}^{n \times m}$ is an $(\epsilon, k)$ projection-cost-preserving sketch for $A$ if*

$$(1 - \epsilon) \|A - PA\|_F^2 \|AS - PAS\|_F^2 \leq (1 + \epsilon) \|A - PA\|_F^2,$$

*for all orthogonal projection matrices $P \in \mathbb{R}^{d \times d}$ of rank at most $k$.*

*Next we check that the composition of a CountSketch and a JL sketch have the PCP property, so that we can apply the same trick as for regression to get truly linear sketching times.*

**Lemma D.7.** *Let $S = S_1 S_2$ where $S_1$ is a CountSketch with small dimension $O(\frac{k^2}{\epsilon^2\delta})$, and $S_2$ is a fast JL transform with small dimension $P(\frac{k}{\epsilon^2} \log \frac{1}{\delta})$. Then with probability at least $1 - O(\delta)$, $S$ is an $(\epsilon, k)$-PCP sketch for a fixed matrix $A$.*

*Proof.* From the definition, the composition of $(\epsilon, k)$-PCP sketches is an $(O(\epsilon), k)$-PCP sketch when $\epsilon < 0.5$. So by Theorem 2 of Musco & Musco (2020), it suffices to check for $i = 1, 2$ that

1. $S_i$ is an $1 \pm O(\epsilon)$ distortion oblivious subspace embedding for $k$-dimensional subspaces with probability $1 - \delta$

---

[5]Recall that a fast JL sketch can be applied to a vector in $\mathbb{R}^n$ in time $O(n \log n)$.

2. $S_i$ has the $O(\frac{\epsilon}{\sqrt{k}})$ approximate matrix multiplication property with probability $1 - \delta$

3. $S_i$ preserves Frobenius norm to $1 \pm O(\epsilon)$ multiplicative error with probability $1 - \delta$

These are all standard facts. For the CountSketch $S_1$, the first property is from Lemma D.1. The second property follows from Lemma 45 of Woodruff et al. (2014), and the third property follows from the JL moment property (see Musco & Musco (2020) for example). The sketching matrix $S_2$ satisfies the $(\epsilon/3, \delta/9^k, 2)$-JL-moment property for the stated size Woodruff et al. (2014), and is immediately a PCP sketch by Musco & Musco (2020). □

*Next we give a version of Theorem 4.1 for estimating Frobenius norm.*

**Lemma D.8.** *For any constant $C \geq 1$ is a $1 \pm \epsilon$ distortion sketch Frobenius estimation sketch $S$ with failure probability $\delta$ where $S$ has small dimension $O(\frac{1}{\epsilon^2} \log \frac{1}{\delta})$ and time $O(C \operatorname{nnz}(x) + d\epsilon^{-2}\delta^{-1/C} \log \frac{1}{\delta})$.*

*Proof.* We use precisely the same sketching matrices as in Theorem 4.1 and almost exactly the same proof.

The minor difference is that sketches need to preserve Frobenius norm for matrices, rather than just $\ell_2$ for a single vector. However, as discussed above, the composition of a CountSketch and a JL sketch yields a sketch satisfying the $(\epsilon, \delta, 2)$-JL-moment property for the dimensions that we choose. The JL moment property is sufficient for Frobenius estimation to $(1 \pm \epsilon)$ multiplicative error with probability $1 - \delta$ [6], so precisely the same proof applies. □

**Theorem D.9.** *Let $A \in \mathbb{R}^{d \times n}$ with $n \gg d$. There is an (oblivious) sketching matrix $S$ such that given $AS$ one can recover a rank $k$ orthogonal projection $\Pi$ such that $\|\Pi A - A\|_F^2 \leq (1+\epsilon) \|A_k - A\|_F^2$, where $A_k$ is the optimal rank $k$ approximation to $A$. Moreover $S$ can be applied in $\operatorname{nnz}(A)$ as long as $\operatorname{nnz}(A) \geq \frac{k^2}{\epsilon^2} d^{1+c_1}$, $S$ succeeds with probability $1 - d^c$ and $S$ has sketching dimension $O(\frac{kd}{\epsilon^2} \log d)$.*

*Proof.* Let $R_1, \ldots, R_T$ all be distributed as the PCP sketch from Lemma D.7 for failure probability $\delta_1$. Note that each $R_i$ can be applied in time $O(\operatorname{nnz}(A) + \frac{dk^2}{\epsilon^2\delta_1} \log(\frac{dk^2}{\epsilon^2\delta_1}))$.

Each $R_i$ is an $(\epsilon, k)$-PCP sketch for $A$ with probability at least $1 - \delta_1$, so with probability at least $1 - \delta_1^T$ at least one of them is. We set $\delta_1 = \delta^{1/T}$ so that at least one of the $R_i$'s is correct with probability at least $1 - \delta$. An $(\epsilon, k)$-PCP sketch for $A$ allows one to output a projection matrix $\Pi$ such that $\|\Pi A - A\|_F^2 \leq (1 + \epsilon) \|A_k - A\|_F^2$. (See Musco & Musco (2020) for details.) Let $\Pi_1, \ldots, \Pi_T$ be the projections recovered from each of our $T$ sketches.

As with regression, we would like to pick the highest quality sketch, so we estimate their errors in parallel (keeping in mind that some PCP sketches might have failed). To do this, we sketch $A$ on the right, using the sketch from Lemma D.8 with distortion parameter $\epsilon$ and failure probability $\delta/T$, so that we can estimate $\|\Pi A - A\|_F^2 \approx \|\Pi A Z_i - A Z_i\|_F^2$. By Lemma D.8, this sketch can be chosen so the sketched matrix has $O(\frac{d}{\epsilon^2} \frac{T}{\delta})$ entries, and can be applied in time $O(C \operatorname{nnz}(A) + d\epsilon^{-2}\delta^{-1/C} \log \frac{1}{\delta})$ for a $C$ that we can choose.

Now we tally the total space and time that we require. The total space for the PCP sketches is $O(T \cdot \frac{kd}{\epsilon^2} \log \frac{1}{\delta})$. All the PCP sketches can be applied in time

$$O(T \operatorname{nnz}(A) + T \frac{dk^2}{\epsilon^2\delta_1} \log(\frac{dk^2}{\epsilon^2\delta_1})) = O(T \operatorname{nnz}(A) + \frac{dk^2}{\epsilon^2\delta^{1/T}} \log(\frac{dk^2}{\epsilon^2\delta})).$$

The total space required for the Frobenius estimation sketch was $O(\frac{d}{\epsilon^2} \frac{T}{\delta})$ and the time required to apply it was

$$O(C \operatorname{nnz}(A) + d\epsilon^{-2}\delta^{-1/C} \log \frac{1}{\delta}).$$

---

[6]For example one way to see this is to note that the JL moment property is preserved under direct sum, as was done in Ahle et al. (2020).

The time to apply the PCP sketches dominates. For $\delta = d^{-c}$ we can choose $C$ to be a constant, so that the total application time is $O(\text{nnz}(A))$ in the parameter regime of the theorem statement.

$\square$

## E    HEAVY-HITTERS RESULTS

**Faster recovery via ExpanderSketch.**    Instead of taking $T_1, \ldots, T_C$ to be arbitrary JL sketches, we can let each $T_i$ be an ExpanderSketch from Larsen et al. (2019). We make a small modification for convenience.

**Proposition E.1.** *There exists a turnstile sketch $T$ with on $\mathbb{R}^d$ with update time $O(\log d)$ using space $O(\epsilon^{-2} \log d)$ along with a recovery algorithm that runs in time $O(\epsilon^{-2} poly \log d)$ and produces a set $\mathcal{H}$ containing the indices of all the $\epsilon$-heavy-hitters and containing only $(\epsilon/3)$-heavy-hitters. The recovery succeeds with probability at least $1 - 1/poly(d)$.*

*Proof.* In parallel, run the ExpanderSketch of Larsen et al. (2019) and an ensemble of CountSketches. The recovery algorithm for ExpanderSketch returns a set $\mathcal{H}'$ of size $O(1/\epsilon^2)$ containing the coordinates of all the $\epsilon$-heavy hitters of $x$. Using the CountSketches we can simultaneously estimate the values of these coordinates to within $(\epsilon/2) \|x\|_2$ additive error, along with $\|x\|$ to within $1 \pm 0.1$ multiplicative error and with failure probability $1/poly(d)$. Then the coordinates with estimated size less than $(\epsilon/2) \|x\|$ may be discarded.    $\square$

As before, we form the sketches $T_i S_i x$. We will then apply the recovery algorithm for the $T_i$'s to obtain the heavy coordinates of $(S_i x)$. Since we know the hash functions used the construct the $S_i$'s, we can then deduce the heavy coordinates of $x$.

We would like to construct our CountSketch distribution $S$ on $\mathbb{R}^{d \times n}$ using hash functions that are easy to invert. For this, assume without loss of generality that $d$ and $n$ are powers of two. We define the hash function family $h_{A,b} : [d] \to [m]$ where $b \in \mathbb{F}_2^{\log m}$ and $A \in \mathbb{F}_2^{\log d \times \log m}$ and $A$ and $b$ are chosen uniformly from the respective spaces. We define

$$h = h_{A,b}(N) = i_m(A\vec{N} + b)$$

where $\vec{N}$ is the $\mathbb{F}_2$ vector corresponding to the binary expansion of $N$, and where $i_m(x)$ gives the number in $[m]$ whose binary expansion is given by $x$. For the CountSketch distribution take $S$ such that $Se_k = \sigma(k)e_{h(k)}$ where $\sigma : [d] \to \{-1, 1\}$ is drawn from any 4-wise independent hash function family.

For completeness, we sketch an argument that this hash function family is indeed pairwise independent.

**Proposition E.2.** *Let $h'_{A,b}(N) = A\vec{N} + b$ as above, where $A$ and $b$ have iid entries over $\mathbb{F}_2$, with the image of $h'_{A,b}$ $\mathbb{R}^M$. Then $h'_{A,b}(x)$ is uniform over $\mathbb{R}^M$ and $h'_{A,b}(x)$ and $h'_{A,b}(y)$ are uniform and independent for $x \neq y$.*

*Proof.* Let $v_1 \neq v_n$ in $\mathbb{F}_2^d$ be arbitrary. We must show that $Av_1 + b$ and $Av_2 + b$ are uniform over pairs of vectors in $\mathbb{R}^M$. Let $f(v) = Av + b$ and note that $f(v_1) - f(v_2) = (Av_1 + b) - (Av_2 + b) = A(v_1 - v_2)$ is uniform over all vectors, since the entries of $A$ are iid. Now note that $f(v_1) - f(v_2)$ is independent of $b$, which is also iid. So conditional on $f(v_1) - f(v_2)$, the quantity $f(v_2)$ is also uniform, and the proposition follows.    $\square$

We recall a standard expected value computation for the buckets of CountSketch.

**Proposition E.3.** *Let $x \in \mathbb{R}^d$ and let $h$ be sampled from a pairwise independently family of hash functions $[d] \to [m]$. Let $\sigma \in \{+1, -1\}^d$ with 4-wise independence among the entries. Let*

$$b_k = \sum_{i:h(i)=k} \sigma_i x_i$$

*be the value in bucket $k$. Then $\mathrm{E}(b_k^2) = \frac{1}{k} \|x\|_2^2$*

*Proof.* Fix, $k$ and let $\chi_i$ be indicator function for the event $h(i) = k$. Then

$$b_k = \sum_i \chi_i x_i \sigma_i.$$

Expanding and applying linearity of expectation gives

$$\mathrm{E}(b_k^2) = \sum_i \mathrm{E}(\chi_i^2 x_i^2 \sigma_i^2) = \frac{1}{k} \|x\|_2^2,$$

since $\mathrm{E}(\sigma_i \sigma_j) = 0$ for $i \neq j$.

$\square$

We also use the fairly standard observation that CountSketch preserves heavy coordinates.

**Proposition E.4.** *(CountSketch maps heavy coordinates to heavy coordinates.) Let $S$ be a CountSketch as detailed above with sketching dimension $m = \Omega(\epsilon^{-2}\delta^{-1})$. Let $x \in \mathbb{R}^d$ and suppose that $|x_k| \geq \epsilon \|x_k\|_2$. Then*

$$\left|(Sx)_{h(k)}\right| \geq (\epsilon/4) \|Sx\|_2$$

*holds with probability at least $1 - \delta$.*

*Proof.* Assume without loss of generality that $\|x\| = 1$. With probability at least $1 - \delta$ we have $\|Sx\|_2 \leq 1 + \epsilon$.

Now conditioned on the value of $h(k)$, the values of $h(i)$ for $i \neq k$ uniform over $[m]$ since our original hash functions were pairwise independent. Let $x_{-k}$ denote $x$ with coordinate $k$ zeroed out. Then we use Markov's inequality to bound the additional mass in bucket $k$. Note that

$$\mathrm{E}((Sx_{-k})^2) = \frac{1}{m} \|x\|_{-k}^2 \leq \frac{1}{m} \leq O(\epsilon^2 \delta).$$

Thus by Markov's inequality, we have that

$$\Pr(|Sx_{-k}| \geq \epsilon/2) = \Pr((Sx_{-k})^2 \geq \frac{\epsilon^2}{4})$$
$$\leq \frac{4}{\epsilon^2} \mathrm{E}((Sx_{-k})^2)$$
$$= \frac{4}{\epsilon^2} \frac{1}{m}$$
$$\leq \delta,$$

as long as we choose $m \geq \frac{1}{4\epsilon^2\delta}$.

Given this, it follows that $\left|(Sx)_{h(k)}\right| \geq \epsilon/2$, so combining with bound on $\|Sx\|_2$ gives

$$\left|(Sx)_{h(k)}\right| / \|Sx\|_2 \geq (\epsilon/2)/(1 + \epsilon) \geq \epsilon/4,$$

with failure probability at most $2\delta$. The desired bound follows by replacing $\delta$ with $\delta/2$.

$\square$

Our approach is to take sketches of the form $TSx$ where $S$ and $T$ are as above. We run the recovery algorithm for $T$ to obtain a set $\mathcal{H}$ containing the heavy hitters of $Sx$. We then let $\mathcal{C}$ be the corresponding set of candidate heavy hitter indices gotten by inverting the hash function $h$. In other words, $\mathcal{C} = h^{-1}(\mathcal{H})$.

**Lemma E.5.** *(i) Suppose that $|x_k| \geq \epsilon \|x\|$. Then with probability at least $1 - (\delta + 1/poly(d))$, $k \in \mathcal{C}$.*

*(ii) Suppose that $|x_k| \leq (\epsilon/10) \|x\|$. Then with probability at least $1 - (\delta + 1/poly(d))$, $k \notin \mathcal{C}$.*

*Proof.* For (i) we have that $\left|(Sx)_{h(k)}\right| \geq (\epsilon/4) \left\|Sx\right\|_2$ with probability at least $1 - \delta$ by the previous claim. The recovery guarantee for $T$ now implies that $h(k)$ occurs in the recovered set $\mathcal{H}$ with probability $1 - 1/\text{poly}(d)$ and hence $k \in \mathcal{C}$.

For (ii) the probability that $\left|(Sx)_{h(k)}\right| \geq \epsilon/3$ is at most $\delta$ by the same Markov bound used above. So the probability that $h(k) \in \mathcal{H}$ is at most $O(\delta + \frac{1}{\text{poly}(d)})$.

$\square$

**Proof of Theorem 4.4**

*Proof.* Take $\delta = d^{-0.1}$ so that the failure probabilities in the above claim can be taken to be $d^{-0.05}$. As long as $\epsilon \geq d^{-0.2}$, then the sketching dimension for a CountSketch $S$ is at most $d^{0.5}$.

Take $C$ CountSketches $S_1, \ldots, S_C$ in parallel with $\mathcal{C}_i$ defined as above. Suppose that $k$ satisfies $|x_k| \leq \epsilon \left\|x\right\|$. By the claim above, the probability that $k$ is in half of the $\mathcal{C}_i$'s is at least

$$1 - \binom{C}{C/2} d^{0.05C/2} \geq 1 - (2d^{-0.025C}),$$

which is at least $1 - 1/\text{poly}(d)$ when $C$ is a sufficiently large constant.

If $|x_k| \leq (\epsilon/10) \left\|x\right\|$ then the probability that $k$ is in at least half of the $\mathcal{C}_i$'s is at most

$$\binom{C}{C/2} d^{-0.05C/2} \leq (2d^{-0.025})^C,$$

which is bounded by an arbitrary inverse polynomial in $d$ by taking $C$ to be a sufficiently large constant.

Hence with all but $1/\text{poly}(d)$ probability, none of coordinates $k$ with $|x_k| \leq (\epsilon/10) \left\|x\right\|$ occur in at least half of the $\mathcal{C}_i$'s. There are only $O(1/\epsilon^2)$ coordinates where this doesn't hold, so at most $O(1/\epsilon^2)$ coordinates $k$ that are in at least half of the $\mathcal{C}_i$'s.

Thus it suffices to find all $k$ that occur in at least half of the $\mathcal{C}_i$'s. To do this we simply take a brute-force approach. There are $\binom{C}{C/2}$ ways to choose a subset $L \subseteq [C]$ indexing half of the $\mathcal{C}_i$'s. For each $i \in L$ there are then $O((1/\epsilon^2)^{C/2}) = \text{poly}(1/\epsilon)$ ways to choose elements $k_1, \ldots k_{C/2}$ from each set in $\{\mathcal{H}_i : i \in L\}$, since each set $\mathcal{H}_i$ has $O(1/\epsilon^2)$ elements. We would like to check if these elements have a common preimage under their respective hash functions. By the construction of our hash functions, this amounts to solving a linear system of dimensions $(C/2 \log d) \times \log m$. Since the sketching dimension $m$ for CountSketch is $O(d)$, this runs in $\text{poly}(\log d)$ time. Repeating for all subsets and choices of $k_1, \ldots, k_{C/2}$ gives a runtime of $O(\binom{C}{C/2}\text{poly}(\log(d)/\epsilon))$. $\square$

### E.1 EXTENSION TO TENSOR SKETCHING.

**Proposition E.6.** *Consider a sketch $S = S_1 \otimes \cdots \otimes S_q$ where each $S_i$ is a CountSketch with hash function $h_i$ down to dimension $m = \frac{c}{q/(\epsilon^2\delta)}$. Let $x$ be a $q$-mode tensor indexed by a multi-index in $[d]^q$, and let the sketch $Sx$ be indexed by a multi-index $[m]^q$. (i) Suppose $i = (i_1, \ldots, i_q)$ is an $\epsilon$-heavy index. Then with probability at least $1 - \delta$, $(h_1(i_1), \ldots, h_q(i_q))$ is $\epsilon/2$-heavy for $Sx$. (ii) Suppose that $i_k$ is not the $k$th coordinate of any $\epsilon$-heavy index. Then with probability at least $1 - O(1/(m\epsilon^2))$, $h_k(i_k)$ is not the first coordinate of any $\epsilon/10$-heavy index of $Sx$.*

*Proof.* The first claim follows from the fact that $S_1 \otimes \cdots \otimes S_q$ has the $(\epsilon, \delta, 2)$-JL-moment property (as we used above). This property implies that point queries estimates $\langle Se_i, Sx \rangle$ approximate $x_i$ to within $\epsilon \left\|x\right\|_2$ additive error with failure probability at most $\delta$. Note that $Se_i = e_{h(i_1), \ldots, h(i_q)}$. We also have that $\left\|Sx\right\|_2 = (1 \pm \epsilon) \left\|x\right\|_2$, and so heavy coordinates remain heavy as desired.

For the second part of the claim, it suffices to prove the claim for the first coordinate. The probability that $h_1(i_1)$ is the $h_1$ hash of the first coordinate of some $\epsilon/5$-heavy-hitter is at most $O(1/(m\epsilon^2))$. So suppose that this does not occur. Then let $\tilde{x}$ be $x$ where all $\epsilon/10$-heavy items are replaced by 0. It suffices to see that $S\tilde{x}$ has no heavy items which follows by the point query bound above. $\square$

**Proof of Theorem 4.7.**

*Proof.* We take our sketch to be of the form $S_1 \otimes S_2 \otimes \cdots \otimes S_q$ where each $S_i$ is a CountSketch with hash function of the form $A\vec{N} + b$. We also repeat this sketch $C$ times independently.

We then compose each of these sketches with the heavy-hitters sketch of Mahankali & Woodruff (2021), which allows us to recover $1/\epsilon^2$ heavy-hitter indices of $S_1 \otimes \cdots \otimes S_q$. We now need to recover the heavy-hitter indices of the original tensor $v$.

We index into $v$ with a multi-index $(i_1, \ldots, i_q)$. To recover the heavy-hitter multi-indices, we first find the indices $i_1$ that correspond to a heavy item. Suppose that $(j_{1k}, \ldots, j_{qk})$ are the multi-indices recovered by the $k$th heavy-hitter sketch. The candidate first indices for the heavy hitters are given by $\bigcup_{k \in [C]} h_1^{-1}(j_{1k})$. With high $1/\text{poly}(n)$ probability, every first coordinate corresponding to some heavy hitter occurs in at least half of the sets in this union. So we may iterate over all $\binom{C}{C/2}$ subsets, and solve the corresponding linear system (it is a linear system due to our choice of hash functions) to find all first coordinates that occur in at least $C/2$ subsets. By the previous proposition there are at most $O(1/\epsilon^2)$ such elements, with high probability as long as $C$ is chosen to be a sufficiently large constant. The same approach applies all other coordinate positions.

Now for each coordinate position, we have a set of $O(1/\epsilon^2)$ candidate heavy-hitter indices. This gives a total of $1/\epsilon^q$ candidate multi-indices. Each of these may be checked directly by applying our point-query sketch from Theorem 4.6. $\square$

# F  $\ell_p$ SAMPLING

The core algorithm for $\ell_p$-sampling reduces the $\ell_p$-sampling problem into a heavy-hitter problem, allowing us to apply our heavy-hitters algorithm as a black box. We first recall the basic setup of the $\ell_p$ sampling algorithms.

**Fact F.1** (Sampling via Exponential Rescaling). *Let $v \in \mathbb{R}^d$. Let $\{E_i\}_{i=1}^d$ be i.i.d. random variables from $Exp(1)$ (an exponential distribution with mean 1). Define a randomly transformed vector $z \in \mathbb{R}^d$ where $z_i = v_i/\sqrt{E_i}$. The index of the maximum magnitude entry of $z$ is a perfect $\ell_2$-sample of $v$:*

$$\Pr\left[i^* = \underset{j \in [d]}{argmax}\{|z_j|\}\right] = \frac{v_i^2}{\|v\|_2^2}$$

*The analysis of [JST11]CITE shows that finding the heavy-hitters of this vector $z$ (for some parameters) provides an approximate $\ell_p$-sample of $v$. Moreover the maximizer $i^*$ is, with good probability, an $(\Omega(1/\log n), \ell_2)$-heavy-hitter of $z$.*

This naturally frames the problem of sampling (and frequency estimation) as a single heavy-hitter problem. The required subroutine is an $\ell_2$-heavy-hitter algorithm that returns both the indices of the heavy items and their estimated values. The fast-application, fast-decode $\ell_2$-HH sketch of Theorem 4.4 does exactly this.

**Theorem F.2** (Linear-Time Sketch for Approximate $\ell_2$-Sampling and Frequency Estimation). *Let $\nu, \delta, \epsilon > 0$ be approximation parameters. There exists a composite linear sketch $S$ such that for any fixed vector $v \in \mathbb{R}^d$:*

1. *The sketch $S \cdot v$ can be computed in $O(d)$ time.*

2. *From the sketch $S \cdot v$, one can compute, in $O(poly(\log(d, 1/\nu, 1/\epsilon)))$ time, a set of index-value pairs $H = \{(i, \tilde{v}_i)\}$. This set constitutes:*

   - *An approximate $\ell_2$-sample of $v$ (i.e., the set $H$ contains all indices $i$ such that $v_i^2 \geq \nu \|v\|_2^2$).*
   - *A $(1 \pm \epsilon)$ relative error frequency estimate $\tilde{v}_i$ for every $i \in H$.*

*This $O(d)$ application time is asymptotically faster than the $O(d \cdot \text{polylog}(d))$ time required to simulate the corresponding streaming algorithm.*

*Proof.* The algorithm follows the standard sketching procedure for $\ell_2$ sampling.

Let $R = \text{diag}(r_1, \ldots, r_d)$ be a diagonal matrix of random variables, where $r_i = 1/\sqrt{E_i}$ and $\{E_i\}$ are i.i.d. draws from $\text{Exp}(1)$. Let $S_{HH}$ be the linear-time, fast-decode $\ell_2$-heavy-hitter sketch. Our composite linear sketch is $S = S_{HH} \circ R$.

**2. Proof of Correctness (Recovery):** The coordinator receives the sketch $A_{full} = SV = S_{HH}(RV)$ where $V = \sum_k v^{(k)}$ is the aggregate vector. Let $Z = RV$. The decoder for $S_{HH}$ is applied to $A_{full}$ and returns the set $H$ of (index, value) pairs $\{(i, \tilde{z}_i)\}$ for the heavy-hitters of $Z$.

- **Sampling:** By Fact F.1 and the analysis in $P_2$, the heavy-hitters of the transformed vector $Z$ correspond to the approximate $\ell_2$-sample of $V$. Thus, the set $H$ is the desired approximate sample.

- **Frequency Estimation:** The same sketch $S_{HH}$ used to find the sample set $H$ also provides the necessary frequency estimates. The decoder for $S_{HH}$ (e.g., ExpanderSketch) returns the estimated value $\tilde{z}_i$ for each $i \in H$. This value satisfies $\tilde{z}_i = (1 \pm \epsilon)z_i$ with high probability.

  The coordinator, having the shared seed for $R$, can compute the scaling factor $r_i = 1/\sqrt{E_i}$ for any $i \in H$. The final frequency estimate is computed by reversing the transform:

  $$\tilde{v}_i = \tilde{z}_i / r_i = (1 \pm \epsilon)z_i \cdot \sqrt{E_i} = (1 \pm \epsilon)(v_i / \sqrt{E_i}) \cdot \sqrt{E_i} = (1 \pm \epsilon)v_i.$$

  This provides the required relative error frequency estimate.

Finally the linear runtime of our sketch follows from the linear runtime of our heavy-hitters sketch.

$\square$

## G  LLM USE DISCLOSURE

LLMs were used for editing purposes and to expand proof outlines.

