# OpenReview forum: "Sketching Faster than Dimension Times Update Time"
_ICLR.cc/2026/Conference — ICLR 2026 Conference Withdrawn Submission_

### Official Review · Reviewer_WB1F · 2025-10-25

**Soundness:** 4
**Presentation:** 2
**Contribution:** 3
**Rating:** 4
**Confidence:** 4

**Summary:**

This paper consider sketching problems with fast update time.  It is not the streaming model, but rather a setting where full dataset, presented as a sequence of updates, is available all at once.  The goal now is to produce
  - asymptotically optimal size sketches
  - that take truly linear time
  - and succeed with high probability.

For instance, a JL sketch can achieve optimal size with high probability, but requires O(1/eps^2) time to process each item read as an update.
Alternatively a CountSketch also has optimal size, and can process updates in truly O(1) update time (e.g., O(nnz(u)) for a sparse vector u, but is not also then with high-probability.

One motivation for this model is a distributed computing setting where data is split among servers which communicate with a single server.  The goal is to sketch the data at each node, and only send the sketch the coordinator.

The paper provides result in this setting for a variety of classic problems in this area:
  - l_2 heavy-hitters
  - F_p moments for p > 2
  - least squared regression (mostly in appendix)
  - oblivious subspace approximation (mostly in appendix)

**Strengths:**

The results combine a lot of advanced machinery from the sketching literature, and chaining these approaches together to get the desired results.


The theoretical results are written with very clearly defined assumptions and rigor.

The model here seems new, and it is possible it could generate further interest.

**Weaknesses:**

There are also often conditions where the error parameter eps must be greater than some quantity (like eps > d^{-0.25)).  These bounds are not so unreasonable, but are somewhat unusual, as it is more common to say eps < 1 or eps < 1/2 for some thing to work, and one can allow eps to approach 0 ... not so here.  This makes me a bit more apprehensive than normal about the utility of the results.

Overall, the results seem rather complex and subtle.  And while it's a theory paper (and that is fine), it feels like it has strayed pretty far from practice to make it less exciting (at least for this reviewer).

Also, the applications of the techniques with the most ties to machine learning (least squares regression, and subspace approximation) are only mentioned in the paper, and discussed mostly in the appendix.  A stronger ICLR submission would have led with these results, and the potential implications of them when analyzed in this model.

**Questions:**

I do not have questions.

---

### Official Review · Reviewer_6z14 · 2025-10-28

**Soundness:** 3
**Presentation:** 2
**Contribution:** 3
**Rating:** 4
**Confidence:** 4

**Summary:**

The paper proposes an algorithm for finding $l_2$ heavy hitters of a vector, i.e a set of indices that exceeds a threshold, that is space optimal and that can be applied in linear time.

In its simplest form the algorithm combines countsketch and JL applied multiple time to boost the probability of success to estimate the $l_2$ norm and the coordinate are than estimated from norms using the identity in line 328 in the paper. In order to obtain a faster recovery and decoding time the paper proposes to combine countsketch + expander sketch instead of JL and obtain a decoding time that is $O(poly(logd))$.

For l_p estimation the paper uses a lower bound on $l_p$ by $l_2$ (line 350) to obtain a linear time sketch with a dimension $\tilde{O}(d^{1-2/p})$.The paper extends the ideas from vectors to tensors and addresses linear regression and low rank approximations and provides algorithms that are linear in time.

The paper discusses the application of these sketching techniques to distributional settings where communication can be done in the sketched space and heavy hitters can be recovered by parties.

**Strengths:**

The paper combines known sketching techniques to obtain algorithms that have desired properties in terms of space and decoding time.
The writing is didactic in parts in the main paper the preliminaries section gives the needed background to read the paper, although it is missing a paragraph on expander sketch. Please add a paragraph here on the expander sketch for unfamiliar readers. Theorems are followed by proof sketches and explanations that put the results in context.

**Weaknesses:**

There are multiple weaknesses in the paper and the appendix. First the paper is missing in the introduction a roadmap concerning the Sections. For instance,  in beginning of Section 3, please guide the reader that this is a summary of main results in the paper, you are also giving same theorems exposed in later sections with different numbers. The paper ends abruptly without any discussion or conclusion.  The appendix is disorganized and does not follow the general structure of the paper. Please start fron l_2 to l_p and map the proofs to the sections in the paper

Another big weakness of the paper is that it relies on repeated applications of the sketch and that the constants in the proof hide constants, without a numerical simulation and experiments it is hard to assess or position the proposed algorithms and to baseline them with respect to competitors in the literature.

For example comparing in wall clock/ and in performance of recovery of heavy hitters of  expander sketch on full vector versus  your countsketch+ expander sketch on some synthetic data. The same goes to the low rank approximation.

Also since you stress a lot on the communication application, a minimal experiment in distributed SGD or in federated learning would also show the viability of this application.

**Questions:**

* If epsilon < d^-0.5 , do you think the sketch would still work and the lower bound you have is an artifact of the proof or this is a fundamental limit?
* The poly term in Theorem 4.4 can it be quantified or measured empirically ?
* Can you add the suggested experiments above ?

---

### Official Review · Reviewer_w56z · 2025-10-31

**Soundness:** 3
**Presentation:** 3
**Contribution:** 2
**Rating:** 4
**Confidence:** 4

**Summary:**

The paper studies the linear sketching problem in the offline setting, where the entire input vector is available in advance rather than arriving as a stream. The key observation is that under this setting, one can compose CountSketch with FastJL or ExpanderSketch to obtain a truly linear-time $O(d)$ algorithm for $\ell_2$-norm estimation, thereby circumventing the known $\omega(1)$ per-update lower bounds from the streaming model. Building on this $\ell_2$ estimation result, the authors further extend their approach to other tasks including $\ell_2$ heavy hitters, $\ell_p$ estimation, linear regression, and low-rank approximation.

**Strengths:**

The paper studies an interesting variant of the sketching problem in a new setting, which is sensible in distributed scenarios. The writing is clear, and the authors successfully extend their main techniques to several classical sketching problems in a natural way.

**Weaknesses:**

1. In distributed settings,people are usually more interested in communication complexity than runtime. It is therefore unclear how revelent this offline setting is.
2. The technical novelty is rather limited: most results follow rather directly from straightforward compositions of existing sketches combined with simple analyses.

**Questions:**

1. If I understand correctly, one might be able to state slightly stronger results: even for the streaming setting, if it's guaranteed that one will only ask for ell_2 estimation/heavy hitters $O(d^{0.01})$ times (as opposed to need to output after each update), you can still get about the exactly same result?
2. I am not fully sure I understand the distinction between the results of Coppersmith and Williams mentioned in Appendix Section C. Is the difference mainly about the number of multiplications versus the overall runtime?

Minor:
the running time of Theorem C.1 should be $O(d^2 poly\log d)$ instead of $O(d poly\log d)$.

---

### Note · Authors · 2026-01-15

I have read and agree with the venue's withdrawal policy on behalf of myself and my co-authors.